# Potential Application of Biochar Composite Derived from Rice Straw and Animal Bones to Improve Plant Growth

**Um-e-Laila, Adnan Hussain, Aisha Nazir \*, Muhammad Shafiq and Firdaus-e-Bareen**

Institute of Botany, University of the Punjab, Lahore 54590, Pakistan; ume.laila1012@gmail.com (U.-e.-L.); hussainadnan268@gmail.com (A.H.); mshafiq.botany@pu.edu.pk (M.S.); firdaus.botany@pu.edu.pk (F.-e.-B.)
\* Correspondence: aisha.botany@pu.edu.pk; Tel.: +92-300-4136566

**Abstract:** The current study is aimed at deriving biochar (BC) from rice straw (RS-BC) and waste bones (WB-BC), being wasted without adequate return at the expense of environmental degradation. The RS and WB feedstocks were pyrolyzed at 550 °C, and the potential of derived biochar as a slow nutrient releasing soil amendment was examined during the growth of ridge gourd. Proximate analysis of the prepared biochars showed significant improvement in ash content and fixed carbon as compared to their raw biomasses. Fourier transform infrared spectroscopy (FTIR), scanning electron microscopy (SEM) and energy dispersive X-ray spectroscopy (EDX) analysis of RS-BC and WB-BC displayed a diverse range of functional groups viz. derivatives of cellulose and hydroxylapatite (HA); macro and microporosity; multiple nutrients. Application of RS-BC and WB-BC in potted soil alone and as biochar composite (RS-BC+WB-BC) at 5, 10 and 15% (*w/w*) and chemical fertilizer (CF) resulted in a significant increase in soil pH, electrical conductivity (ECe), cation exchange capacity (CEC) and water holding capacity (WHC) in exchange for growth and yield of ridge gourd. However, there were insignificant differences in the growth of plants in response to RS-BC, WB-BC alone and CF with biochar composite at 15% amendment. For giving insignificantly different growth results than CF, the prepared biochar composite showed outstanding potential as an organic fertilizer applicable in agrarian soils to elevate soil properties and yield of agricultural commodities.

**Keywords:** animal bones; bone char; pyrolysis; rice straw; slow-release fertilizer

## 1. Introduction

The production of waste in the form of rice straw and animal waste including bones is on the verge of increasing with increasing demand for rice, chicken and meat in populous countries, such as Pakistan. According to the report of economic survey, Pakistan is ranked second in good quality rice production in the world with 9.76 Mt rice residues, brings out a yield of more than 7 million tons and ranked 10th in the world with respect to the production rate [1]. Conventionally, the rice straw is burnt in the fields, which adversely influences the environmental quality and releases greenhouse gases such as CO, $CO_2$, $CH_4$ and $NO_X$ into the atmosphere, causes the formation of smog affecting human health [2]. At the same time, millions of tons of chicken, mutton and beef bones are being produced annually. Animal bones have a massive amount of nutrients, minerals and proteins which are often part of municipal solid waste [3].

Production of biochar seems an appropriate way to manage the voluminous quantity of crop residues including rice straw. Most of the times, residues of many of the commonly cultivated crops such as, rice, etc., are either burnt on-field or used as poor fuels in the remote villages of low-income countries. Pyrolysis is thermal degradation of organic biomass under a restricted supply of oxygen with temperature ranging from 300–600 °C [4]. The ultimate output of pyrolysis is biochar, syngas and bio-oil. The yield of pyrolysis depends upon the various factors associated with this pyrolysis, such as pyrolysis condition, temperature and time of processing [5]. Currently, biochar has achieved worldwide

attention due to its carbon sequestration technology and its sustainable usage in the field of agriculture [6]. Dong et al. [7] reported that biochar acts as SRF (slow-release fertilizer), which builds up the nutrient accessibility to the soil very slowly and diminishes the loss of nutrients through leaching. So, biochar not only acts as a soil conditioner but also improves crop yield. Hossain et al. [8] reviewed biochar as a value-added product that can be used as a fertilizer. Biochar use in soil improves its pH, EC, nutrient accessibility and high-water retention capability due to having unique morphological and chemical features, such as charge on surface, surface area and surface porousness. The morphological and physico-chemical features are mainly impacted by the kind of feedstock, heating grade and conditions provided during pyrolysis [9,10].

Population explosion has necessitated the demand for agricultural production at a fast pace. The use of chemical fertilizers has increased to a great extent to obtain high crop yield to fulfill the demand of communities [11]. Fertilizers used to increase crop yield affect the physical condition of the soil by altering soil acidity, microbial behavior and organic matter content [12]. Furthermore, there has always been a price hike of the chemical fertilizers that severely affect the small and medium farmers of the low-income countries. Deriving biochar from crop residue and wastes for application in vegetable crops could be an economically advantageous approach.

Many articles have been published so far regarding rice straw-derived biochar production along with its characterization and sorption response [13] for climate change mitigation, remediation of pollutants, restriction and degradation of contaminants [14]. Previously, animal bone char has been used as an organic fertilizer due to having an excessive amount of Ca and P [15]. At the same time, some researchers have worked on the photocatalytic activity of bone char and its implication as a supportive material for the production of nanoparticles used for cleaning the environment [16]. Xiao et al. [17] worked on the macro and micro nitrogenous bone char attained by using the moistened milling technique, and their results indicated that it can be used for the decontamination of the aquatic environment.

The effect of two waste materials, rice straw and bones as biochar composite on the characteristics of soil and crop productivity have scarcely been investigated. Their use as a biochar composite to improve soil characteristics and to enhance the crop productivity through provision of macro and micronutrients needs to be investigated. The current research was aimed at studying individual and combined effects of RS-BC, WB-BC against CF and control on physico-chemical and biological properties of soil with subsequent effects on the growth and yield of ridge gourd (*Luffa acutangula* (L.) Roxb.).

## 2. Materials and Methods

### 2.1. Collection of Biomass

Rice straw was collected from three random sites of a pre-irrigated field area of University of the Punjab, Lahore, Pakistan (location: latitude 31°35′30″ N; longitude 074°17′30″ E). However, the waste bones, including chicken, beef and mutton bones, were scavenged from solid waste dumping sites of hostels and canteens of University of the Punjab.

### 2.2. Preparation of Feedstock

The collected biomasses were washed with water and air dried. Both biomasses were chopped into small pieces of uniform size (2 cm) with the help of an electric cutter. The prepared feedstock of uniform size was used for pyrolysis.

### 2.3. Preparation of Biochar

The pre-weighed feedstock was fed into the feedstock chamber of the pyrolyzing unit. Biochar was prepared from rice straw (RS) and waste bones separately at a temperature of 550 °C for a residence time of 25 min under extremely inadequate provision of oxygen i.e., by achieving 0.001 Torr pressure in the feedstock chamber of a locally assembled

semi-automatic pyrolyser with ramp rate of 30 °C min$^{-1}$. Each prepared biochar was stored in polythene bags prior to use. Yield of each prepared biochar was calculated by using the following formula.

$$Yield\ (\%) = W_{BC}/W_{RB} \times 100 \tag{1}$$

where $W_{RB}$ is the mass of raw material and $W_{BC}$ is the mass of prepared biochar.

### 2.4. Proximate and Ultimate Analysis

Proximate analysis of raw biomass and prepared biochar were carried out to ascertain the moisture content (MC), volatile content (VC) and ash content (AC) following the standard procedure [18].

1. $MC\ (\%\ w/w) = a - b/a$
2. $VC\ (\%\ w/w) = b - c/b$
3. $AC\ (\%\ w/w) = c - d/c$

where a is the weight of air-dried sample, b is the weight of sample dried at 105 °C for 24 h, c is the weight of sample dried at 950 °C for 6 min, d is the weight of sample dried at 750 °C for 6 h. However, fixed carbon (FC) was detected by subtracting all the above proximate parameters from 100 as given in Equation (4).

4. $FC\ (\%) = 100 - (MC\ (\%) + VC(\%) + AC\ (\%))$

The elemental analysis of C,H,N,S of raw biomass and prepared biochar were performed by using an elemental analyzer system (GmbH, Vario MICRO cube V1.9.4) but oxygen was determined by subtracting the mass, while considering C,H,N and ash content of the entire mass.

### 2.5. Physico-Chemical Analysis

The pH, electrical conductivity ($EC_e$) and total dissolved solids (TDS) of the derived biochar were determined by mixing one gram of the sample in 20 mL of distilled water. The suspension was left for 3–4 h, the solution was mixed well and filtered. The pH, $EC_e$ and TDS of the filtrate was determined [19]. As given by Brewer and Levine [20], bulk density (BD) was calculated while taking the 100 mL of cylinder filled with ground biochar sample after tapping it at least three times to obtain a persistent volume.

$$BD = ground\ biochar/packed\ volume\ of\ biochar$$

Cation exchange capacity (CEC) of the prepared biochar was calculated by using an ammonium acetate solution by following the protocol given by Chapman [21]. For estimation of nutrients one gram of biochar sample was digested in $HClO_4$: $HNO_3$ (4:1 ratio) The digested solution was cleared through heating and the final volume made up to 100 mL using distilled water [22].

### 2.6. SEM, EDX and FTIR Analysis

Biochar surface morphology was revealed through scanning electron microscopy (SEM, JEOL JSM-6480LV) by putting the biochar sample on a two-edge carbon tape, sticking to aluminum stub. Energy dispersive X-ray spectroscopy (EDX) was performed to detect the quantitative values of numerous elements. Functional groups present in the RS biochar and bone char samples were observed by using Fourier Transformation Infrared Spectroscopy (FTIR) by using the spectrophotometer (IR Prestige-21 Shimadzu, Japan). Biochar samples were analyzed between the wavelength ranges of 400–4000 cm$^{-1}$ at the effective resolution of 4 cm$^{-1}$. Results were examined by means of spectrum IR (solution software).

### 2.7. Soil Sampling

The soil used for experiments was collected from a field with no reported history of heavy metal contamination. The physico-chemical properties of the air dried, sieved and thoroughly mixed soil were determined by using standard procedures.

### 2.8. Experimental Set Up

The experiment was conducted in a randomized complete block design to lessen the confounding errors due to location. Each treatment was replicated thrice. Pots possessing the capacity of 2.5 kg of soil were selected for the experiment. The treatments along with control comprised:

- Control soil (no biochar amendment, no fertilizer).
- Rice straw-derived biochar (RS-BC) at three levels of amendment: soil amended with 5% biochar (5% RS-BC, *w/w*), soil amended with 10% biochar (10% RS-BC, *w/w*), soil amended with 15% biochar (15% RS-BC, *w/w*).
- Bone char at three levels of amendment: soil amended with 5% bone char (5% WB-BC, *w/w*), soil amended with 10% biochar (10% WB-BC, *w/w*), soil amended with 15% biochar (15% WB-BC, *w/w*).
- Biochar composite prepared by combination of rice straw-derived biochar and bone char with 1:1 ratio at three levels: soil amended with 5% biochar (5% RS-BC+WB-BC, *w/w*), soil amended with 10% biochar (10% RS-BC+WB-BC, *w/w*), soil mixed with 15% biochar (15% RS-BC+WB-BC, *w/w*).
- Commercial fertilizer (NPK, recommended dose for ridge gourd).

Before planting each potted soil supplemented with biochar, it was moistened at around 60% moisture content and left in the greenhouse for a pre-incubation period of 15 days to enhance the chemical reaction between the biochar and soil. In this experiment, the ridge gourd (*Luffa acutangula* L. Roxb.) belonging to the family Cucurbitaceae was used as a test plant because of its high biomass production and quick flowering. Five healthy germinated seeds of ridge gourd were transferred to each pot. Pots were watered on alternate days to keep them at pot capacity during the whole growth period. The plants were allowed to grow for 65 days under the temperature of $27 \pm 2$ °C in a greenhouse.

### 2.9. Soil Analysis

Pre-planting and post-harvest soil analysis was carried out physico-chemically. For this purpose, soil cores were taken from each potted soil, the collected soil samples were air dried, crushed in pestle and mortar and sieved through using a <2 mm sieve. Soil pH, $EC_e$ and TDS were determined in soil extract (1:1). Soil organic matter (OM) was detected by following loss on ignition method. Total organic carbon was determined according to the method of Walkley [23], bulk density was calculated as above. The cation exchange capacity was determined by following the method of Chapman [21]. Mineral constitution of each soil amendment was assessed by acid digestion of soil sample followed by atomic absorption spectrometry [22]. However, water holding capacity of pot soil was determined by using the method given by Gessert [24].

### 2.10. Determination of Growth and Yield

The effects of various treatments on plant height, plant dry biomass, number of fruits (luffa) and chlorophyll content (SPAD values) were determined at the point of harvest of 65 days old plants.

### 2.11. Statistical Analysis

The data obtained from this study were statistically analyzed by using ANOVA 9.3 version to calculate the means and standard deviation. All the treatment means were compared by using the least significance difference ($p = 0.05$).

## 3. Results and Discussion

### 3.1. Characterization of Feedstocks and Derived Biochars

The biochar yield was 32% and 39.8% for rice straw biochar (RS-BC) and bone char (WB-BC), respectively, (Table 1) at high pyrolysis temperature of 550 °C. As a matter of fact, it was reported that the increase in pyrolysis temperature from 300 °C to 600 °C could adversely influence the yield of rice straw biochar due to the thermal degradation of massive hydrocarbons [2]. Krzesinska and Majewska [25] showed that pyrolysis of bovine bones resulted in the 30% yield of bone char.

Results of proximate analysis are given in Table 1, illustrating that raw biomass initially had more moisture content and volatile content of about 21.12% and 26.09% for RS and 24.75% and 22.06% for WB biomass, respectively. Moreover, ash content and fixed carbon were found to be very low in raw biomasses compared to their derived biochar, which were primarily 20.39% and 32.52% in RS and 36.25% and 17% for WB, respectively (Table 1). Biomass that possesses ash contents greater than or substantially equivalent to 15 is considered valuable substances to be used for pyrolysis [26]. After thermal decomposition of raw biomasses, there was a significant decrease in volatile content of both prepared biochars. It was confirmed by Li et al. [27] that when biomass is subjected to high temperature it results in the reduction of biochar yield and loss of volatile contents. The percentage of ash contents and fixed carbon were increased to 42.39% and 49.04% for RS-BC and 30% and 59.9% for WB-BC, respectively. This increase in ash content and fixed carbon might be the recalcitrant carbon coming into existence due to the accumulation of carbon and minerals content at high pyrolysis temperature. Cao and Harris [28] also indicated that when pyrolysis temperature expands, ash content also increases because of a significant rise in quantity of organic residues and minerals despite the loss of volatiles.

**Table 1.** Proximate and ultimate analysis of rice straw (RS), waste bone (WB) feedstocks and their derived biochars.

|  |  | **RS** | **WB** | **RS-BC** | **WB-BC** |
|---|---|---|---|---|---|
| **Biochar yield (%)** |  | - | - | $32.0^B \pm 2.28$ | $39.8^A \pm 3.01$ |
| **Proximate analysis (wt.%)** | MC | $21^B \pm 2.23$ | $24^A \pm 3.56$ | $6^D \pm 0.66$ | $8^C \pm 0.89$ |
|  | VC | $26^A \pm 3.44$ | $22^B \pm 1.88$ | $2^C \pm 0.03$ | $1^{CD} \pm 0.02$ |
|  | AC | $20^C \pm 2.08$ | $17^D \pm 2.06$ | $42^A \pm 4.02$ | $30^B \pm 3.03$ |
|  | FC | $32^D \pm 2.88$ | $36^C \pm 2.09$ | $49^B \pm 4.02$ | $59^A \pm 5.06$ |
| **Ultimate analysis (wt.%)** | C | $44^B \pm 3.55$ | $12^D \pm 1.50$ | $52^A \pm 5.01$ | $28^C \pm 2.44$ |
|  | H | $5^{BC} \pm 1.88$ | $7^A \pm 2.02$ | $2^D \pm 0.04$ | $5^B \pm 0.67$ |
|  | N | $2^{CD} \pm 0.02$ | $10^A \pm 0.02$ | $3^C \pm 0.08$ | $8^B \pm 1.86$ |
|  | O | $28^B \pm 3.22$ | $53^A \pm 6.02$ | $11^C \pm 0.02$ | $27^B \pm 2.84$ |
| **Ratio** | C/H | $8^B \pm 1.18$ | $1^D \pm 0.06$ | $24^A \pm 2.68$ | $5^{BC} \pm 1.02$ |
|  | C/N | $21^A \pm 2.88$ | $1^D \pm 0.05$ | $15^B \pm 1.69$ | $3^C \pm 0.22$ |

**Note:** MC—moisture content, VC—volatile contents, AC—ash contents, FC—fixed carbon, C—carbon, H—hydrogen, N—nitrogen, O—oxygen. Values are means of three replicates ± SD. The values represented with different letters are significantly different with reference to Duncan multiple range test ($p$ = 0.05).

Physico-chemical properties of the prepared rice straw biochar and bone char are shown in Table 2. The results indicated that, as compared to raw feedstocks, values of pH and $EC_e$ of biochars were significantly higher. pH of rice straw was increased to 9.52 in rice straw-derived biochar. At the same time, pH was found to be highly alkaline (9.4) in bone char. Similarly, the $EC_e$ of both biochar samples were significantly enhanced to 454 µS cm$^{-1}$ and 1370 µS cm$^{-1}$ for rice straw biochar and bone char, respectively. Moreover, the CEC of bone char was higher than that that of RS-derived biochar. The nutrient levels revealed that RS-BC and WB-BC have a massive amount of macro and microelements such as Na, N, K, P, Ca, Mg, Zn and Mn, as given in Table 2. RS-BC comprised K (0.085 g kg$^{-1}$) and N (0.095 g kg$^{-1}$) and higher quantity of micronutrients such as Cu (0.053 g kg$^{-1}$), Zn (0.058 g kg$^{-1}$) and Mn (0.035 g kg$^{-1}$). Bone char showed the

highest value of macronutrients such as 0.095 g kg$^{-1}$ Ca, 0.075 g kg$^{-1}$ P and 0.043 g kg$^{-1}$ of Mg as supported by the EDX analysis. These observations were consistent with the findings of preceding researchers, indicating that biochar consists of remarkable amounts of C, H, N and lower quantities of other mineral nutrients [29–31]. In current study, carbon content of 52.2% and 28.4% was found for RS-BC and WB-BC, respectively, which can contribute as a source and substrate for microbial activity in soil. The C/N ratio of RS-BC was obtained lower; however, it was found higher in WB-BC as compared to feedstock, as given in Table 1. The reduced H, N and O depicting the hydrophobicity might be due to the loss of volatile organics during heat treatment in line with the findings reported by Rafiq et al. [32] and Singh et al. [13].

**Table 2.** Physico-chemical analysis of rice straw-derived biochar (RS-BC) and bone char (WB-BC) prepared at 550 °C.

| Samples | Physico-Chemical Properties | | | | Nutrients (g kg$^{-1}$) | | | | | | | | |
| | pH | EC$_e$ (µS cm$^{-1}$) | CEC (cmol kg$^{-1}$) | BD (g cm$^{-3}$) | Na | K | Ca | Mg | N | P | Cu | Zn | Mn |
|---|---|---|---|---|---|---|---|---|---|---|---|---|---|
| **RS-BC** | 9.52 ± 0.82 | 454 ± 8.64 | 6.80 ± 0.48 | 0.23 ± 0.06 | 0.013 ± 0.01 | 0.085 ± 0.02 | 0.019 ± 0.01 | 0.014 ± 0.01 | 0.095 ± 0.05 | 0.038 ± 0.02 | 0.053 ± 0.03 | 0.058 ± 0.04 | 0.035 ± 0.02 |
| **WB-BC** | 9.40 ± 0.78 | 1370 ± 9.92 | 7.08 ± 0.39 | 0.44 ± 0.02 | 0.048 ± 0.03 | 0.042 ± 0.02 | 0.095 ± 0.04 | 0.043 ± 0.01 | 0.038 ± 0.02 | 0.075 ± 0.02 | 0.025 ± 0.01 | 0.032 ± 0.02 | 0.028 ± 0.01 |

**Note:** ECe—electrical conductivity of biochar extract, CEC—cation exchange capacity, BD—bulk density, Na—sodium, K—potassium, Ca—calcium, Mg—magnesium, N—nitrogen, P—phosphorus, Cu—copper, Zn—zinc, Mn—manganese. Values are means of three replicates ± SD.

### 3.2. SEM, EDX and FTIR Analysis

Scanning Electron Microscopic images of rice straw biochar and bone char at different magnifications are shown in Figure 1. At low magnification, the smooth structure of biochar of rice straw biochar was observed as compared to high magnification indicating a rough structure having a pore size ranging from 1.21 µm to 1.52 µm, clearly visible along the vascular bundles. The pores fall in the class of macropores as set out by the International Union of Applied Chemistry, the pore size of >50 nm in width [33]. However, surface morphology of the prepared bone char indicated less porosity. Instead, agglomeration of dense constituents represented its bulky structure at high magnification of 2500x (B). Rice straw biochar showed a smooth, porous and tubular structure, which reinforces that it can hold more nutrients [13]. Animal bones under high thermal treatment of 600 lessened the bulk porosity of obtained bone char. Similar evidences by Krzesińska and Majewska [24] indicate that thermal degradation of animal bones at high temperature results in the stiffness of bone char surface morphology.

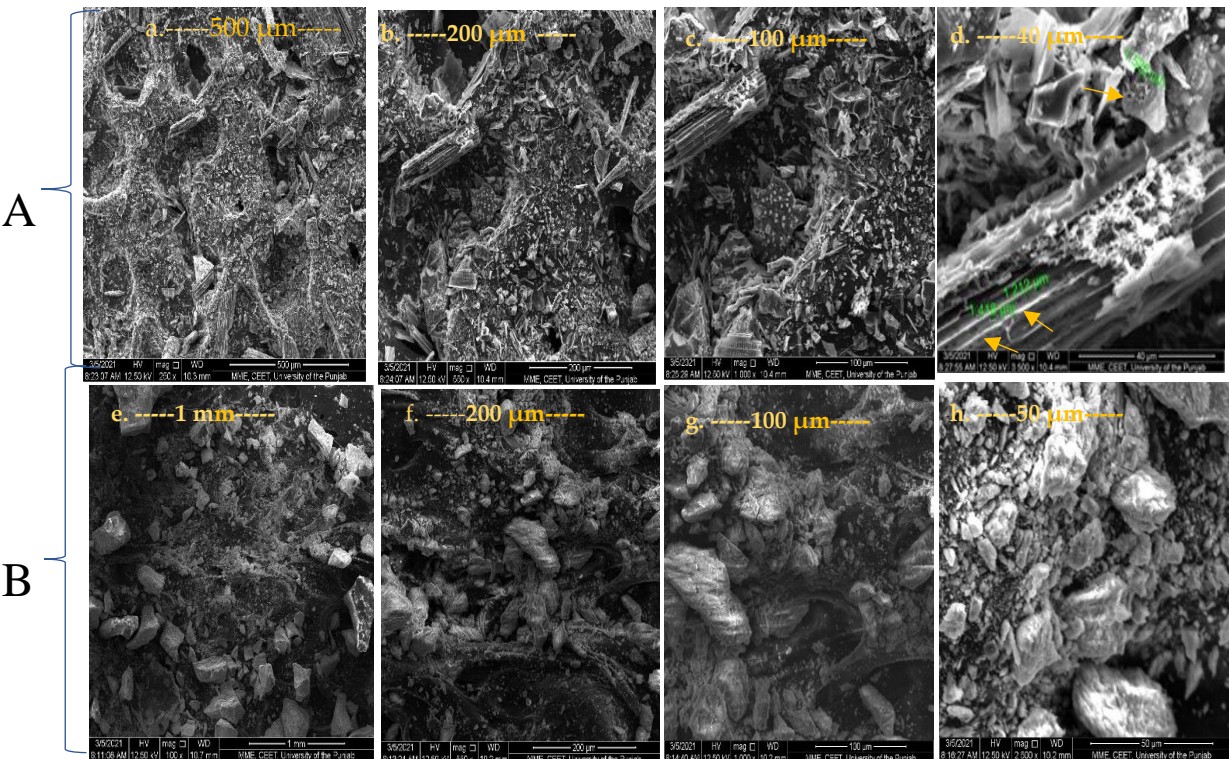

**Figure 1.** Scanning electron micrographs of (**A**) rice straw-derived biochar; (**B**) bone char with different morphological Figure (**a**) 250×, (**b**) 550×, (**c**) 1000×, (**d**) 2500×, (**e**) 100×, (**f**) 550×, (**g**) 1000×, and (**h**) 2500× magnifications.

Energy dispersive X-ray spectroscopy of RS-BC and WB-BC was carried out to confirm the presence of major elements (Figure 2). EDX spectrum indicated that in rice straw biochar, the main constituents such as C, K, Si and O with atomic weights of 63.83, 5.288, 30.60 and 5.40%, respectively, were found, as shown in Table 3. A promising value of 11.08 for C/K was observed. Ca, P and O, which are considered as key shares of hydroxyapatite, were observed in bone char with high percentages. The Ca/P ratio was found to be 2.02 in bone char, which was substantially equivalent to value 1.9 + 0.5, as presented by Akindoyo et al. [34]. Lower value of Ca/P might be due to the alteration of amorphous structure of calcium and phosphorous [35].

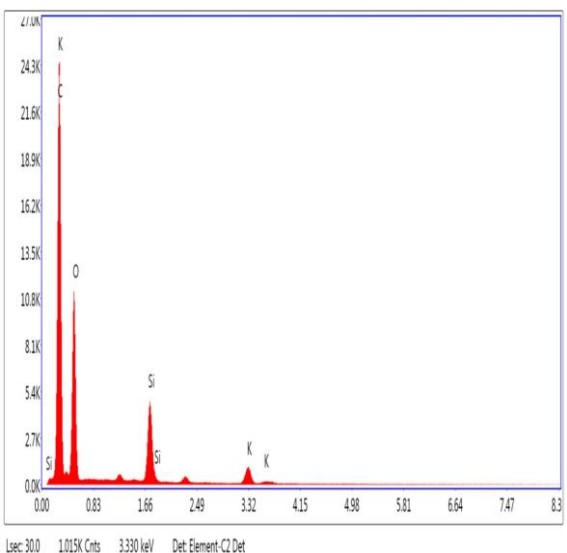

(a)

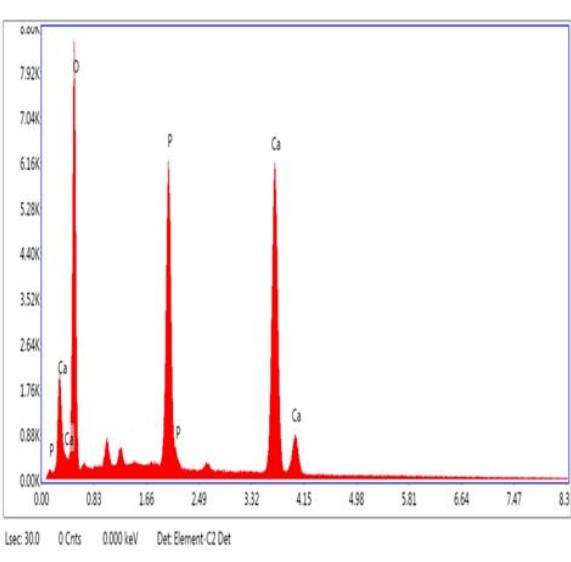

(b)

**Figure 2.** EDX spectra of (**a**) rice straw-derived biochar; (**b**) bone char.

**Table 3.** EDX data attained from the elemental analysis of rice straw-derived biochar (RS-BC) and bone char (WB-BC).

| Samples | C | | K | | Si | | O | | |
|---|---|---|---|---|---|---|---|---|---|
| **RS-BC** | Weight % | Atomic % | Weight % | Atomic % | Weight % | Atomic % | Weight % | Atomic % | C/K Ratio |
| | $53.62 \pm 4.44$ | $63.83 \pm 5.12$ | $10.31 \pm 0.56$ | $5.28 \pm 0.23$ | $34.24 \pm 2.66$ | $30.60 \pm 2.18$ | $10.2 \pm 0.49$ | $5.40 \pm 0.38$ | $11.08 \pm 0.76$ |
| | Ca | | P | | O | | | | |
| **WB-BC** | Weight % | Atomic % | Weight % | Atomic % | Weight % | Atomic % | | Ca/P Ratio | |
| | $43.77 \pm 3.02$ | $26.64 \pm 1.88$ | $16.75 \pm 1.01$ | $13.19 \pm 0.88$ | $39.48 \pm 4.22$ | $60.18 \pm 6.01$ | | $2.02 \pm 0.03$ | |

Note: C—carbon, K—potassium, O—oxygen, Si—silicon, Ca—calcium, P —phosphorus, O—oxygen. Values are means of three replicates ± SD.

FTIR analysis indicated a diverse range of functional groups (Figure 3). Alkali functional groups are the indication of high level of pH and $EC_e$. Peaks observed during the analysis were at different wavelengths ranging from 4000–400 cm$^{-1}$. For both RS-BC and

WB-BC, medium strong peaks were detected at 3854 cm$^{-1}$, 3740 cm$^{-1}$, 3690 cm$^{-1}$, specifying the OH stretching vibrations which are the characteristic functional group for cellulose and HAP (Hydroxyapatite) [36]. Around 3500 cm$^{-1}$, medium intense peaks were observed illustrating the N-H stretch for amine groups present in the RS-BC and WB-BC samples. The peak at 2351 cm$^{-1}$ indicated the strong $O = C = O$ stretch in RS-BC represented by the presence of carbon dioxide, which was not clearly visible in WB-BC. At 1990 cm$^{-1}$, the $N = C = S$ stretch was observed in RS-BC, whereas such a peak disappeared in the WB-BC sample. It represents the isothiocyanate that is mainly obtained from the plant source. At 1750 cm$^{-1}$, the $C = O$ stretch represents the esters group, while 1550 cm$^{-1}$ and 1500 cm$^{-1}$ represents the $C = O$ and $C = C$ stretch for RS-BC and WB-BC samples. Akindoyo et al. [34] analyzed the IR spectrum of the prepared cow bone char and found peaks at 1411 and 1457 cm$^{-1}$, which indicates the presence of $CO_3{}^{-2}$ lattice in the HAP. Wu et al. [37] and Peng et al. [38] showed similar groups in RS-BC at wavenumber of 800–1600 cm$^{-1}$. The curve obtained at about 1052–1050 cm$^{-1}$ showed the CO and CO-O-CO stretches, illustrating the presence of anhydride groups in prepared biochars. Some of the curves obtained at <900 cm$^{-1}$ showed presence of Si-O/Si-C structures.

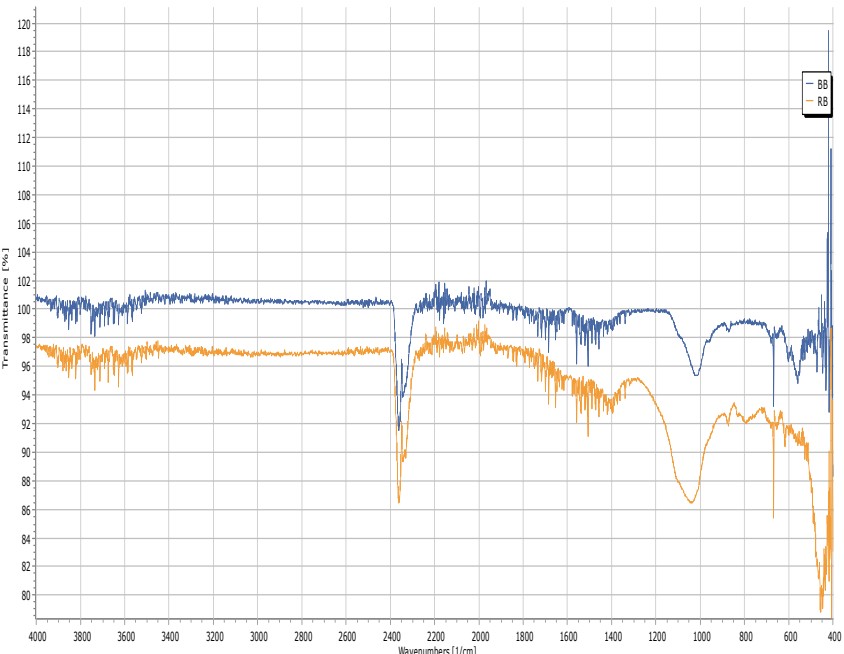

**Figure 3.** FTIR spectra of rice straw-derived biochar and bone char.

### 3.3. Improvement of Soil Supplemented with RS-BC and WB-BC

The pH, EC$_e$, organic matter content, water holding capacity, CEC and soil organic carbon significantly increased in biochar amended soil (Table 4). However, the application of RS-BC+WB-BC was found more effective. The pH and EC$_e$ of soil were recorded to be 8.9 and 171.73 at 15% RS-BC+WB-BC application. The organic carbon attained from biochar changed soil properties while enhancing activity of microbes and pH of the soil [39]. Soil pH is considered an important factor that directly influences the microbe's activity and transferring of nutrients [40]. The soil organic carbon also increased significantly with the increasing dose of RS-BC and this enhancement occurred due to increase in the C/N ratio and addition of nutrients in the biochar amended soil. The results are in agreement with the findings of Chen et al. [15], showing that animal-based biochar poses a greater influence on soil pH than the plant-based biochar. CEC represents the bulk of cations associated to the biochar surface. The CEC of soil tended to increase effectively after the addition of biochar due to the presence of the functional group and negative charge on the surface of biochar [41]. The results were similar to the findings of Peng et al. [38] who reported that RS-derived biochar significantly raised the pH value by 0.1 to 0.46 units in soil and

increased the CEC of soil by 17%. Most importantly, it was also observed that the bulk density of soil after biochar addition decreased significantly indicating the fact that added biochar is porous in nature. The changes in physical properties of soil may also provide space for microbial activity, which also promotes the defense mechanism for plants [42,43]. Mixing biochar to the soil reduced the soil bulk density (BD) because biochar is light in weight and its density is significantly lower than that of soil [44]. Normally, the reduced bulk density of soil results in the provision of more macro- and less micro-pore space for the retention of more moisture and soil nutrients.

**Table 4.** Physico-chemical characteristics of soil having different application rates of rice straw-derived biochar (RS-BC) and bone char (WB-BC) alone and in combination (RS-BC+WB-BC) after a 15-day incubation period. Values are means of three replicates ± SD.

| Treatments | Application Rate | Soil Parameters | | | | | | | |
|---|---|---|---|---|---|---|---|---|---|
| | | pH | ECe ($\mu S\ cm^{-1}$) | OM (%) | WHC (%) | TDS (ppm) | CEC ($cmol_c\ kg^{-1}$) | BD ($g\ cm^{-3}$) | SOC (%) |
| Control | | $7.95^{CD} \pm 0.10$ | $104^{EF} \pm 6.30$ | $3.13^{F} \pm 0.03$ | $28^{D} \pm 5.68$ | $118^{CD} \pm 16.99$ | $8.33^{E} \pm 0.29$ | $1.40^{B} \pm 0.05$ | $3.56^{CD} \pm 0.20$ |
| CF | | $8.19^{BCD} \pm 0.04$ | $161^{B} \pm 3.90$ | $3.48^{E} \pm 0.04$ | $27^{D} \pm 4.04$ | $139^{BC} \pm 6.55$ | $9.2^{E} \pm 0.38$ | $1.58^{A} \pm 0.11$ | $4.35^{BC} \pm 0.41$ |
| RS-BC | 5% | $7.60^{cE} \pm 0.22$ | $108^{cE} \pm 6.10$ | $3.41^{cE} \pm 0.11$ | $32^{cC} \pm 6.24$ | $135^{aBC} \pm 6.62$ | $9.6^{bE} \pm 0.35$ | $1.28^{aCD} \pm 0.02$ | $4.73^{cB} \pm 0.17$ |
| | 10% | $7.93^{bCD} \pm 0.1$ | $114^{bD} \pm 5.86$ | $3.59^{bD} \pm 0.15$ | $55^{aB} \pm 8.02$ | $108^{bD} \pm 3.15$ | $11^{abD} \pm 0.57$ | $1.14^{bDE} \pm 0.03$ | $5.30^{bB} \pm 0.17$ |
| | 15% | $8.25^{aBC} \pm 0.1$ | $131^{aC} \pm 3.23$ | $3.94^{aCD} \pm 0.08$ | $67^{aA} \pm 2.64$ | $145^{bD} \pm 10.25$ | $12.5^{aC} \pm 0.37$ | $1.02^{cE} \pm 0.02$ | $6.26^{aA} \pm 0.37$ |
| WB-BC | 5% | $7.9^{bDE} \pm 0.32$ | $109^{cE} \pm 12.09$ | $3.64^{bD} \pm 0.20$ | $30^{bCD} \pm 6.02$ | $81^{cE} \pm 7.91$ | $11.1^{cD} \pm 0.18$ | $1.35^{aC} \pm 0.04$ | $3.16^{bD} \pm 0.11$ |
| | 10% | $8.1^{bBCD} \pm 0.17$ | $117^{bD} \pm 20.12$ | $3.98^{bBC} \pm 0.12$ | $50^{aBC} \pm 7.7$ | $109^{bD} \pm 18.36$ | $12.9^{bBC} \pm 0.52$ | $1.24^{bCD} \pm 0.03$ | $4.58^{aBC} \pm 1.69$ |
| | 15% | $8.4^{aB} \pm 0.18$ | $183^{aA} \pm 23.23$ | $4.40^{aA} \pm 0.47$ | $63^{aAB} \pm 6.8$ | $169^{aA} \pm 12.09$ | $14.6^{aA} \pm 0.44$ | $1.12^{cDE} \pm 0.02$ | $4.73^{aB} \pm 0.18$ |
| RS-BC+WB-BC | 5% | $8.12^{cBCD} \pm 0.1$ | $124^{cCD} \pm 8.67$ | $3.93^{bCD} \pm 0.20$ | $34^{bC} \pm 3.05$ | $146^{aB} \pm 17.77$ | $11.9^{bCD} \pm 0.54$ | $1.36^{aC} \pm 0.04$ | $4.99^{aB} \pm 0.18$ |
| | 10% | $8.42^{bB} \pm 0.09$ | $127^{bC} \pm 9.32$ | $4.22^{abBC} \pm 0.1$ | $53^{abB} \pm 5.10$ | $107^{bD} \pm 7.47$ | $14^{aAB} \pm 0.57$ | $1.28^{bCD} \pm 0.02$ | $4.89^{aB} \pm 0.22$ |
| | 15% | $8.9^{aA} \pm 0.22$ | $171^{aA} \pm 9.86$ | $4.33^{aA} \pm 0.20$ | $66^{aA} \pm 3.21$ | $136^{aBC} \pm 12.61$ | $15.33^{aA} \pm 0.4$ | $1.21^{bD} \pm 0.01$ | $5.24^{aB} \pm 0.19$ |

**Note:** ECe—electrical conductivity of soil extract, OM—organic matter, WHC—water holding capacity, TDS—total dissolved solids, CEC—cation exchange capacity, BD—bulk density, SOC—soil organic carbon. The statistical differences are represented by an upper-case letter among the treatments and lower-case letter within each treatment. The values represented with different letters are significantly different with reference to Duncan multiple range test ($p = 0.05$).

### 3.4. Improvement in Post Harvested Soil

After harvesting the ridge gourd, the values obtained from WB-BC and RS-BC+WB-BC treated soils were significantly improved as compared to control and soil applied with commercial fertilizer (Table 5). The biochar composite of RS-BC+WB-BC acted as a slow-release organic fertilizer, significantly improving soil properties, as given by Koron et al. [45] who reported that biochar-derived from plant sources has less impact on soil fertility as compared to that of bone char which comprised about 90% mineral composition and 10% carbon content. Chen et al. [15] also reported that the addition of bone char increased carbon mineralization in soil and thus can be utilized as an organic fertilizer because it has the property of hydroxyapatite and an adequate amount of Ca and P contents. During an experimental study, Butnan et al. [46] clarified that adding biochar in soil boosts up the soil properties, such as WHC, SOC, CEC, and improves nutrient availability in soil. Our results also support that biochar as a supplement elevated soil pH resulting in soil alkalization, which caused a rise in ECe and CEC of the soil. It is assumed that the application of RS-BC and WB-BC in the form of biochar composites induced soil microbial activities, which made the macro and micronutrients available to the plants, most likely through the process of soil microbial solubilization [47]. Thus biochar-soil microbes build a strong relationship with plant by increasing nutrient availability.

**Table 5.** Physico-chemical characteristics of post-harvested soil after harvest of 65-day-old ridge gourd plants. Values are means of three replicates ± SD.

| Treatments | Application rate | pH | ECe ($\mu$S cm$^{-1}$) | OM (%) | WHC (%) | TDS (ppm) | CEC (cmol$_c$ kg$^{-1}$) | BD (g cm$^{-3}$) | SOC (%) |
|---|---|---|---|---|---|---|---|---|---|
| Control | | 7.55$^C$ ± 0.05 | 118$^H$ ± 2.73 | 3.74$^{CD}$ ± 0.11 | 28$^F$ ± 1.04 | 126$^H$ ± 3.68 | 7.9$^G$ ± 0.16 | 1.38$^A$ ± 0.01 | 3.47$^F$ ± 0.07 |
| CF | | 8.1$^C$ ± 0.06 | 155$^E$ ± 4.98 | 3.75$^{CD}$ ± 0.05 | 30$^E$ ± 2.51 | 140$^G$ ± 4.46 | 8.8$^{FG}$ ± 25 | 1.32$^{AB}$ ± 0.02 | 4.67$^E$ ± 0.15 |
| RS-BC | 5% | 7.62$^{cD}$ ± 0.10 | 123$^{cG}$ ± 2.56 | 3.64$^{bD}$ ± 0.10 | 35$^{cD}$ ± 1.15 | 139$^{bG}$ ± 3.07 | 9.5$^{bF}$ ± 0.34 | 1.14$^{aBC}$ ± 0.01 | 6.43$^{cD}$ ± 1.28 |
| | 10% | 7.94$^{bC}$ ± 0.10 | 130$^{bF}$ ± 3.81 | 3.77$^{abCD}$ ± 0.04 | 38$^{bD}$ ± 2.64 | 145$^{bFG}$ ± 4.75 | 10$^{abE}$ ± 0.52 | 1.12$^{bFG}$ ± 0.02 | 6.56$^{bCD}$ ± 1.25 |
| | 15% | 8.21$^{aBC}$ ± 0.07 | 145$^{aF}$ ± 3.2 | 3.92$^{aC}$ ± 0.08 | 58$^{aC}$ ± 3.51 | 171$^{aE}$ ± 6.50 | 12.3$^{aCD}$ ± 0.35 | 1.10$^{cC}$ ± 0.01 | 7.81$^{aC}$ ± 0.54 |
| WB-BC | 5% | 7.56$^{bD}$ ± 0.18 | 159$^{cE}$ ± 4.04 | 3.98$^{bC}$ ± 0.23 | 56$^{bC}$ ± 1.52 | 126$^{cH}$ ± 4.85 | 11$^{bDE}$ ± 0.31 | 1.21$^{aB}$ ± 0.03 | 6.48$^{cD}$ ± 1.28 |
| | 10% | 8.20$^{aBC}$ ± 0.05 | 171$^{bD}$ ± 3.77 | 4.65$^{aB}$ ± 0.15 | 62$^{aB}$ ± 2.61 | 153$^{bF}$ ± 5.93 | 12.80$^{bC}$ ± 0.5 | 1.18$^{bB}$ ± 0.01 | 8.01$^{bB}$ ± 0.57 |
| | 15% | 8.21$^{aBC}$ ± 0.03 | 199$^{aC}$ ± 5.21 | 4.71$^{aB}$ ± 0.18 | 63$^{aBC}$ ± 0.57 | 189$^{aD}$ ± 5.01 | 15$^{aAB}$ ± 0.23 | 1.16$^{cBC}$ ± 0.02 | 8.58$^{aA}$ ± 0.38 |
| RS-BC+WB-BC | 5% | 8.43$^{aAB}$ ± 0.07 | 241$^{cB}$ ± 5.38 | 4.69$^{bB}$ ± 0.12 | 63$^{bBC}$ ± 1.52 | 226$^{cC}$ ± 9.40 | 12$^{bCDE}$ ± 0.31 | 1.18$^{aB}$ ± 0.03 | 7.81$^{cC}$ ± 0.17 |
| | 10% | 8.49$^{aA}$ ± 0.39 | 246$^{bB}$ ± 15.86 | 4.84$^{bB}$ ± 0.08 | 69$^{aAB}$ ± 0.43 | 315$^{bA}$ ± 10 | 14.6$^{aB}$ ± 0.30 | 1.08$^{bCD}$ ± 0.01 | 8.08$^{bB}$ ± 0.52 |
| | 15% | 8.67$^{aA}$ ± 0.09 | 340$^{aA}$ ± 5.68 | 5.26$^{aA}$ ± 0.25 | 74$^{aA}$ ± 0.57 | 291$^{aB}$ ± 4.04 | 16.1$^{aA}$ ± 0.21 | 1.03$^{bD}$ ± 0.02 | 8.64$^{aA}$ ± 0.25 |

Note: The statistical differences are represented by an upper-case letter among the treatments and lower-case letter within each treatment. The values represented with different letters are statistically different with reference to Duncan multiple range test ($p$ = 0.05).

### 3.5. Improvement in Growth and Yield of Plants Grown in Biochar Amended Soil

Results obtained from crop growth and yield parameters were statistically compared within and along each treatment to ensure that biochar treatments at different levels improved crop productivity or not (Figure 4). The growth and yield of ridge gourd tended to be significantly enhanced under all biochar application levels, either applied individually or in the combination as compared to control. One very interesting observation was that plants cultivated in biochar amended soils took only 45 days to reach flowering stage, which was 10 days earlier than that of control, and the highest male and female flowers were observed under the treatment of biochar composite during the growth period. Perhaps, this could be related to the organic volatiles deposited on the surface of biochar, which could have triggered hormonal changes in the plants that had induced early flowering in the ridge gourd. This implication, however, requires future detailed studies potentially by applying the extracts of fresh biochar for elucidating their effects on plants as triggering source of hormonal change. The ascending order of soil amendments for all the growth parameters, especially plant dry biomass, was as: RS-BC+WB-BC>WB-BC>RS-BC>CF; however, no significant improvement was found in fruit yield. However, fruit weight was significantly higher (59%) in biochar composite (RS-BC+WB-BC) amended soil as compared to control (Table 6) with dry biomass of plants being insignificantly higher than biomass obtained in CF i.e., the biochar composite gave ridge gourd yield as equal as CF. Similar findings have been reported in previous studies [48,49]. The chlorophyll contents (SPAD values) in the ridge gourd leaves were significantly higher in the soils applied with individual biochar and biochar composite as compared to control and CF. It was clearly noticed that when biochar composite (RS-BC+WB-BC) was used, all growth parameters increased significantly with an increasing biochar application rate. One of the possible reasons could be that the biochar composite provided a diverse and greater number of biochar surfaces with the distribution of electrostatic charge on its surfaces that could have led to a slow release of macro and micronutrients (as illustrated in the EDX spectrum given in Figure 2) in soil. Ultimately, this could have led to significant improvement in the growth of the plant due to low nutrient loss as compared to CF, where due to the presence of the least number of and uniform type of soil colloid surfaces might have caused leaching of nutrients and reduced nutrient accessibility to the plants. In the current study, biochar treated soil (15% $w/w$) resulted in a significant influence on the accumulation of chlorophyll content, which might be due to the bioavailability of slowly released nutrients viz. N, Mg and P release from biochar, responsible for the photosynthetic activity in plants. Nonetheless, N and Mg are recognized as the integral factors for the accumulation of chlorophyll content in plant leaves. That was the obvious reason that chlorophyll content increased with the increasing level of biochar amendments. Such kind

of findings have been reported by Libutti et al. [50]. On an average, the highest value of 100 cm was recorded at the application level of 15% RS-BC+WB-BC. Azeem et al. [51] have shown that the application of cow bone at the rate of 2.5% improved the soil properties and plant attributes, including the chlorophyll content; however, the application rate of 10% (*w/w*) bone char reduced the soil enzymatic activities and ultimately diminished the plant growth. During the current study, promising growth results of ridge gourd were obtained possibly due to a huge quantity of essential nutrients (N, P, K, Ca and Si) in the WB-BC with sufficient bioavailability in the soil. It was quite likely that biochar application had incurred a striking impact on the morphological and biochemical parameters of the ridge gourd as it had shown comparatively more growth in soils amended with biochar than either of control or CF (Figure 4). As highlighted by Liu et al. [52], the RS-derived biochar enhances soil organic carbon and nitrogen use efficiency; the improvement of crop yield in biochar amended soil compared to commercial fertilizer could possibly be because of the biochar addition. Furthermore, improvement in plant growth could be due to bioavailability of soil nutrients in synergism with the elevated soil physico-chemical properties rendered by the biochar, as described in other studies [53,54].

**Table 6.** Variation in fruit weight (g) of *Luffa acutangula* L. Roxb. cultivated under the different application level of rice straw-derived biochar (RS-BC), bone char (WB-BC) and biochar composite (RS-BC+WB-BC) along the control and commercial fertilizer (CF). Values are means of three replicates±SD.

| Parameters. | Control | CF | RS-BC | | | WB-BC | | | RS-BC+WB-BC | | |
|---|---|---|---|---|---|---|---|---|---|---|---|
| | | | 5% | 10% | 15% | 5% | 10% | 15% | 5% | 10% | 15% |
| Fruit weight (g) | $143^{H} \pm 2.33$ | $328^{A} \pm 4.44$ | $192^{F} \pm 2.64$ | $220^{E} \pm 2.66$ | $155^{G} \pm 2.34$ | $280^{D} \pm 2.82$ | $302^{C} \pm 4.68$ | $303^{C} \pm 3.55$ | $301^{C} \pm 4.88$ | $305^{B} \pm 5.88$ | $336^{A} \pm 6.33$ |

Note. The values represented with different letters are statistically different with reference to Duncan multiple range test (*p* = 0.05).

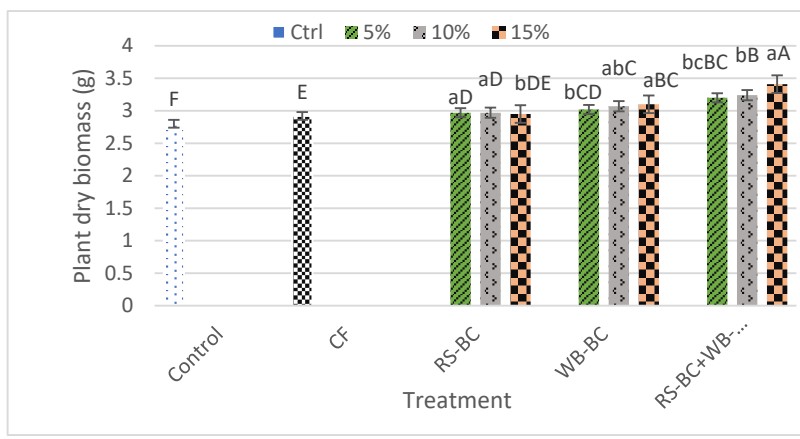

**a**

**Figure 4.** *Cont.*

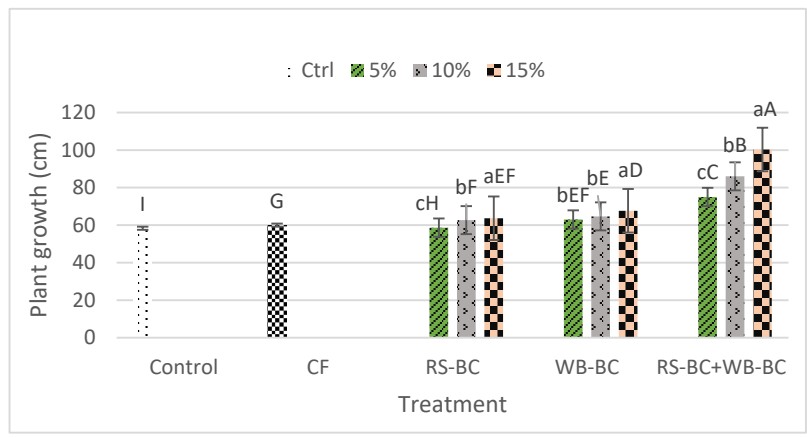

**b**

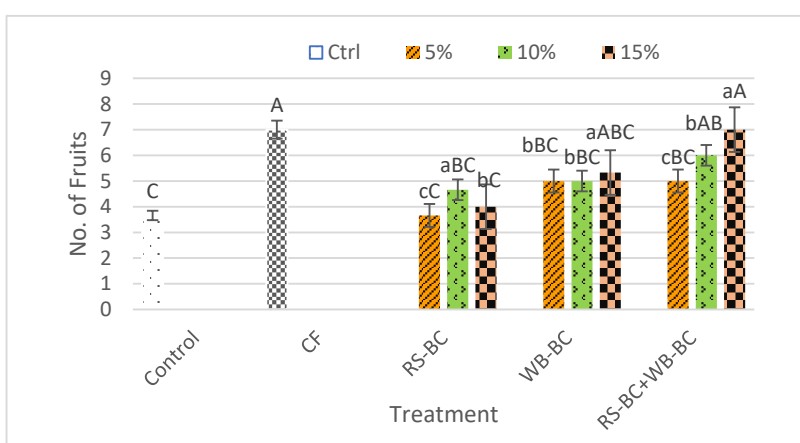

**c**

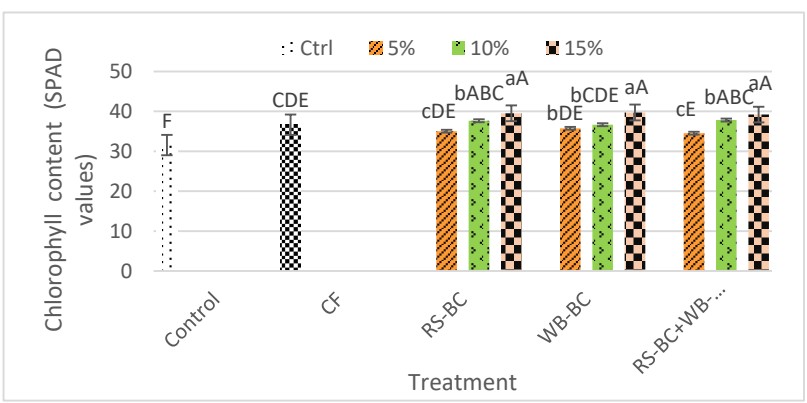

**d**

**Figure 4.** Variation in growth attributes: (**a**) plant height; (**b**) dry biomass; (**c**) number of fruits; (**d**) SPAD values of *Luffa acutangula* L. Roxb cultivated under the different application level of rice straw-derived biochar (RS-BC), bone char (WB-BC) and biochar composite (RS-BC+WB-BC) along the control and commercial fertilizer (CF). The statistical differences are represented by an upper-case letter among the treatments and lower-case letter within each treatment. The values represented with different letters are statistically different with reference to Duncan multiple range test ($p$ = 0.05).

## 4. Conclusions

The discharge of huge amounts of recyclable bio-waste, particularly rice straw and waste bones, can be put to effective use by preparing its biochar. The addition of such composite biochar in soil resulted in earlier flowering and more fruit yield than commercial fertilizer, thus reducing the application of chemical fertilizers. The application of RS-BC+WB-BC biochar composite had incremented growth of ridge gourd than each of the individually applied RS-BC and WB-BC, being co-translated by significant improvement in the physico-chemical and biological properties of biochar amended soils. The WB-BC performed better than RS-BC as an organic soil amendment. Future research can be performed on the effect of fresh and aged biochar derived from feedstocks with high organic volatile contents, such as WB-BC in terms of its triggering effect on beneficial plant hormonal changes.

**Author Contributions:** Conceptualization, A.N. and M.S.; methodology, U.-e.-L., A.H., A.N. and M.S.; validation, M.S.; formal analysis, U.-e.-L.; investigation, A.H., resources, M.S. and F.-e.-B.; data curation, A.N. and M.S.; writing, original draft preparation, U.-e.-L. and A.N.; writing, review and editing, U.-e.-L., A.N., M.S., and F.-e.-B.; supervision, A.N. and F.-e.-B.; funding acquisition, F.-e.-B. All authors have read and agreed to the published version of the manuscript.

**Funding:** The current study was solely funded by University of the Punjab, Pakistan under the PURC Research Projects from "Lumpsum provision for Research" for the Financial Year 2020-21 (Project No. 08 and 09 vide letter No. Sr. D. 1700/Est-I, Diary No. 9162-AAI).

**Data Availability Statement:** Delivered on demand.

**Acknowledgments:** Not applicable.

**Conflicts of Interest:** The authors declare no conflict of interest.

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
