# Peer review of "Potential Application of Biochar Composite Derived from Rice Straw and Animal Bones to Improve Plant Growth"

_sustainability, doi:10.3390/su131911104_

Round 1
Reviewer 1 Report
The manuscript entitled "Potential application of biochar composite derived from rice straw and animal bones to improve plant growth" is interesting to read. It is dealing with a contemporary topic of waste management and its utilization as soil ameliorants. Though the topic and work is highly researched nowadays. The manuscript has some merit and can be publishable after revisions suggested in the attached pdf file.
Specifically, language needs attention as there are some words which are not correctly framed at different places. The authors have chosen rice- straw and bone-biomass for biochar preparation which are generally a common issue to manage in Pakistan. Authors need to throw some light on global importance of these feedstocks. Moreover, the plant chosen for the study should be described for its global/local significance. As there are a number of studies on rice-straw biochar and its impact on different soil and plant parameters inlcluding cereal and legume plants. Also authors need to clearly establish why these two biochars were mixed, i.e. logic behind it (improving nutrient content/soil CEC, etc) in the introduction section. Well, manuscript is presented well and can be considered after highlighting the novelty of the work with reference to previous literature.

Author Response
Please find following attachments:
- Response of Authors to the Reviewer-1's comments
- Edited manuscript file with track changes

Reviewer 2 Report
The authors should do corrections. Following are the suggestions and changes
- Please add more reference in the introduction especially for statements like
- “Animal bones have a massive amount of nutrients, minerals, and proteins which are often part of municipal solid waste.”
- “According to the report of economic survey, Pakistan is ranked second in good quality rice production in the world, brings out a yield of more than 7 million tons and ranked 10th in the world with respect to the production rate.”
- Improper usage in the introduction 2nd paragraph “Dong et al. Dong et al. [6] reported that biochar acts as SRF (slow-release fertilizer) which build up the nutrient accessibility to the soil very slowly and diminishes the loss of nutrients through leaching.”
- Incomplete statement in
- Introduction 4th paragraph “ Previously, animal bone char has been used as organic fertilizer due to having excessive amount of Ca and [14].”
- Section 2.5 “Cation exchange capacity (CEC) of the prepared biochar was calculated by using ammonium acetate solution by following the protocol given by [19].”
- In Table 2, Electrical conductivity is denoted as EC while in rest of the text it is mentioned as ECe. Please check and correct it. If they are different , what is the difference. Similarly in section 3.3 first line…”The pH, EC, organic matter content” this is visible
- In section 2.4, for calculating FC, “However, fixed carbon (FC) was detected by subtracting all the above proximate parameters from 100 as given in Equation 3
- FC (%) = 100 − ( MC (%)+ VC(%)+ AC (%))
Please check the equation number
- Add a reference in section 2.5 bulk density calculation part
- In section 2.7 authors have mentioned they used heavy metal-free soil for plant study but did not provide soil analysis results before planting
- Rephrase statement in section 2.9 “Total organic carbon [21], while bulk density was calculated as above and cation exchange capacity was determined by following the method of Chapman [19].”
- Section 2.10 “The effects of various treatments on plant height, plant dry biomass, no of fruits and chlorophyll content (SPAD values) were determined at the end of growth period”.
What does author mean by “number of fruits.”, this is a vegetable growth being carried out?
- In section 3.1, “The nutrient levels revealed that RS-BC and WB-BC have a massive amount of macro and microelements such as Na, N, K, P, Ca, Mg, Zn and Mn Table 2.”
Please write Table 2 within brackets or change it to “as shown in Table 2.”
Similar mistake in section 3.4 “After harvesting the ridge gourd, the values obtained from WB-BC and RS-BC+WB-BC treated soils were significantly improved as compared to control soil and soil with commercial fertilizer Table 5.”
Section 3.2 “Energy Dispersive X-ray Spectroscopy of rice straw biochar and bone char was carried out to confirm the presence of major elements Figure 2”
- In post harvested soil, from Table 5, RS-BC addition of 15% decreases SOC. What may be the possible reason?
- The authors should modifiy the section 3.3 and 3.4 and include maximum information rather talking about one parameter in plant growth studies. The authors should utilize the data in Figure 4 to provide the explanations to the different results obtained.
- In the conclusion part should be re-written. The authors have not mentioned about which biochar composite s good, no information about its effect on plant growth studies.
- The reference format is not correct in the 21st
Author Response
Please find following attachments:
- Response of Authors to the comments of Reviewer 2
- Edited manuscript with track changes

Reviewer 3 Report
My main concern about this paper is the lack of context in the Introduction. I think that if authors amend this point, the paper can be published. My recommendation for the Introduction:
- Authors begin the Intro explaining about waste production in Pakistan, but they do not compare with other countries, so the reader cannot know if the amount of waste is big or slow.
- After that they go directly to explain that the production of biochar is the best way. Nonetheless, for novel readers, authors must describe what others thermochemical treatments are available and which are the advantages of biochar production in contrast with these other technologies.
- Authors propose directly the use of biochar to improve plant growth. Nonetheless, biochar could be also used as fossil carbon substitute. Why is better to use it as plant growth composite? The only reason relays on economic purposes, but authors have not mentioned it. Please see: doi.org/10.1016/j.scitotenv.2019.06.484 ;doi.org/10.1016/j.scitotenv.2021.147169; doi.org/10.1007/s12649-012-9190-y
- Finally, most emphasis needs to be put on the aim and objectives of the paper.
- Conclusion also must contains further future works
Author Response
Please find following attachments:
- Response of Authors to the Comments of Reviewer 3
- Edited manuscript with track changes

Round 2
Reviewer 1 Report
The manuscript has been revised considerably in light of the previous comments. However, authors have not attended most of the comments/suggestions/queries related to the data of the study. I have some queries related to the validity of the data, especially related to the soil properties after amendment.
Please see the attached file and thoroughly revised the manuscript.

Author Response
Point by point response to the comments of the Reviewer 1 (Round 2)
Comment: flow is missing. In the previous paragraph rice straw burning is mentioned!
Response: flow is developed between the preceding and following paragraph.
Comment: N2 purging was done? elaborate the pyrolysis condition as it is the major determinants of biochar properties!
Response: Instead of using N2 purging, the deairing of the feedstock chamber was carried out by achieving high vacuum of 0.001 Torr in the feedstock chamber.
Comment: recheck the formula! It should be like Wbc/Wrb x 100. I already suggested in my previous review comments!
Response: The formula is corrected as suggested.
Comment: how can you find such soil as heavy metals are also the components of soil mass and most of the soil contain inherent amount of Fe, Al, Ti, etc. Rephrase for clarity or elaborate it!
Response: The sentence has rephrased for required clarity.
Comment: Soil samples were homogeneous for each treatment? as the data table reflect some discrepancy in the soil samples.
Response: The soil samples were made homogenous by mixing in the V-mixer after sieving biochar and soil through < 2 mm sieve. After that, the soil was filled in the pots. Likewise, after harvesting the plants the soil was again crushed and passed through <2 mm sieve.
Comment: mention the method for determining water holding capacity?
Response: The required method has been provided.
Comment: at the end? means after drying? rephrase for clarity!
Response: The sentence is rephrased to add required clarity.
Comment: no need to write this sentence, it can be merged in the previous one.
Response: The two sentences are merged into one statement.
Comment: These data are based on AAS analysis or EDX analysis? As EDX analysis provides the crude estimate of the percent composition, one cannot directly mention it as the accurate amount!
Response: The given values of the metals were based on AAS analysis.
Comment: support with a few more studies!
Response: Further studies have been added as suggested.
Comment: based on EDX analysis? you can only provide tentative estimates and presence of different minerals!
Response: The given values of the biochar nutrients were determined by AAS.
Comment: some pics of Si-O/ Si-C are also there (<900 cm-1).
Response: The presence of Si-O/ Si-C at <900 cm-1 have been described in the Results.
Comment: 10 days? as mentioned in the material and methods section? Check and correct accordingly!
Response: The 10 days have been corrected to 15 days in the 2.8 Experimental set up.
Comment: the control and CF soils would be same during the initial days? I doubt the data! It might be due to heterogeneity of the soil systems!
Response: The data typos in the derived Table 4 have been rectified after verification from the MS Excel source data file.
Comment: it should be close to the control soil samples! There is some heterogeneity in the samples!
Response: The data typos rectified after verification.
Comment: drastic change in the physical properties in just 50 days? Even the soil ameliorative effects of biochar on different physical properties have been drastically reduced and equilibrates to the control soils as compared to that observed during the initial days! Recheck the values!
Response: Although total days were 80 (15 days of equating of the soil amendments + 65 days of plant growth) the data typos have been rectified after verification.
Reviewer 2 Report
The manuscript is recommended for publication
Author Response
The Reviewer 2 has recommended the manuscript for publication without any further comments.
Round 3
Reviewer 1 Report
The authors have make some changes as highlighted in my previous two review comments. However, there are still several issues which need authors attention. Specifically, I am not sure about the data presented in this study which needs major attention of the authors. It seems that authors have written this manuscript in hurry as there are several methodological/calculation corrections which have been corrected while the previous reviews and there are still many to be corrected. This reflect the less clarity/soundness of the study findings in its present form. I am wondering to see the changes in some of the parameters after biochar amendment in this 65 days old experiment. Authors need to substantially support/discuss such variations in light of relevant literature. Authors may see the attached file for more details. I truly respect editors' remarks for this manuscript.

Author Response
Dear Editor
Thank you very much for your deep analysis of the manuscript sustainability-1360665 and valuable comments. The authors have very carefully addressed the comments of Reviewer 1 given during Review Round 3. The point-by-point response of the authors to the comments of the Reviewer 1 during Review Round 3 are given as under, as well as, provided in the attached pdf file. The updated manuscript file with track changes have been duly submitted on the Journal submission portal.
We hope that the revised manuscript will satisfy the Reviewers and will fulfill the high standards of your well reputed Journal.
Sincerely,
Dr Aisha Nazir
Corresponding Author (Manuscript ID: sustainability-1360665)
Round 3: Response to Reviewer 1 Comments
Comment 1: Abbreviation should be elaborated at first appearance for ECe, CEC and WHC.
Response 1: Abbreviation has been elaborated as electrical conductivity of extract (ECe), cation exchange capacity (CEC) and water holding capacity (WHC)
Comment 2: “outstanding potential as a balanced biofertilizer” overambitious statement, rephrase it for clarity.
Response 2: Rephrased it as “outstanding potential as an organic fertilizer”
Comment 3: Auspicious charge
Response 3: The word ‘auspicious’ has been deleted. However, auspicious charge was meant to be promising and significantly higher charge on surface as compared to the control i.e. unpyrolyzed feedstock.
Comment 4: ramp rate of 30°C s -1. Correct it
Response 4: Corrected as 30°C min-1
Comment 5: (5% RS-BC w/w) correct it
Response 5: Typo corrected as suggested (10% RS-BC w/w).
Comment 6: Table 1 dots are misplaced in some values; data are not properly presented.
Response 6: Dots are inserted once again and corrected the data in Table 1, as suggested by the reviewer.
Table 1. Proximate and ultimate analysis of rice straw (RS), waste bone (WB) feedstocks and their derived biochars
|
|
|
RS |
WB |
RS-BC |
WB-BC |
|
Biochar yield (%) |
|
- |
- |
32.0B±2.28 |
39.8A±3.01 |
|
Proximate analysis (wt.%) |
MC VC AC FC |
21.12B±2.23 26.09A±3.44 20.39C±2.08 32.52D±2.88 |
24.75A±3.56 22.06B±1.88 17.05D±2.06 36.25C±2.09 |
6.57D±0.66 2.01C±0.03 42.39A±4.02 49.04B±4.02 |
8.57C±0.89 1.50CD±0.02 30.02B±3.03 59.9A±5.06 |
|
Ultimate analysis (wt.%) |
C H N O C/H C/N |
44.2B±3.55 5.10BC±1.88 2.09CD±0.02 28.22B±3.22 8.67B±1.18 21.14A±2.88 |
12.11D±1.50 7.52A±2.02 10.22A±0.02 53.1A±6.02 1.61D±0.06 1.18D±0.05 |
52.2A±5.01 2.11D±0.04 3.28C±0.08 11.2C±0.02 24.74A±2.68 15.91B±1.69 |
28.4C±2.44 5.26B±0.67 8.88B±1.86 27.4B±2.84 5.39BC±1.02 3.20C±0.22 |
|
Ratio |
Comment 7: pH of was, correct the sentence
Response 7: removed “of” in the sentence.
Comment 8: refer the appropriate Table/Figure while explaining C/N ratio
Response 8: The description of C/N ratio of the biochar is referred to Table 1.
Comment 9: Admin highlighted the word “found”.
Response 9: removed it from the sentence and corrected it.
Comment 10: this enhancement occurred due to increase in C/N ratio and addition of nutrients of in amended soil. Rephrase it for clarity.
Response 10: Rephrased it as ‘this enhancement occurred due to increase in C/N ratio and concentration of nutrients in biochar amended soil.’
Comment 11: Admin highlighted “diminish” word.
Response 11: the word ‘diminish’ replaced with “reduced” the BD of soil.
Comment 12: In table. 4 and table 5. Data presented seems to be varying in some parameters. Recheck the data and CEC unit.
Response 12: During the data handling, values were misplaced by the typist, we carefully recalculated and examined the data. Now this has been corrected as presented below in Table 4 and Table 5. CEC unit has been corrected as cmolc kg-1.
The pH, ECe and TDS was determined by following method given by Bordoloi et al. [19].
Comment 13: Generally N fertilizer application increases the leaf N and chlorophyll content. Authors should clearly explain it why biochar applied soils resulted in high chlorophyll content.
Response 13: Yes this is right generally N fertilizer application increases the leaf N and chlorophyll content but in current study, biochar treated soil (15% w/w) resulted significant influence on accumulation of chlorophyll content which might be due to the bioavailability of nutrients viz. N, Mg and P release from biochar, responsible for the photosynthetic activity in plants. Nonetheless, N and Mg are recognized as the integral factors for the accumulation of chlorophyll content in plant leaves. That was the obvious reason that chlorophyll content increased with the increasing level of biochar amendments. Such kind of findings have been reported in the literature e.g. Libutti et al. [48].
- Libutti, A.; Trotta, V.; Rivelli, A. R. Biochar, Vermicompost, and Compost as Soil Organic Amendments: Influence on Growth Parameters, Nitrate and Chlorophyll Content of Swiss Chard (Beta vulgaris L. var. cycla). Agron. 2020, 10(3), 346. https://doi.org/10.3390/agronomy10030346.
Comment 14: Discussion part needs more improvement to elaborate the changes in the light of recent literature.
Response 14: Elaborated some key changes in light of recent literature.
Liu et al. (50) highlighted that RS-derived biochar had a great potential in enhancement of soil organic carbon and nitrogen use efficiency favored the improvement of crop yield as compared to NPK fertilizer. Furthermore, the improvement in plant growth can be due the bioavailability of nutrients and associated elevated soil physico-chemical properties including WHC and porosity as described by others El-Naggar et al. [51]; Ghosh et al. [52].
- Liu, J.; Jiang, B.; Shen, J.; Zhu, X.; Yi, W.; Li, Y.; Wu, J. Contrasting effects of straw and straw-derived biochar applications on soil carbon accumulation and nitrogen use efficiency in double-rice cropping systems. Agric. Ecosyst. Environ. 2021, 311, 107286. https://doi.org/10.1016/j.agee.2020.107286.
- El-Naggar, A.; Lee, S. S.; Rinklebe, J.; Farooq, M.; Song, H.; Sarmah, A. K.; Ok, Y. S. Biochar application to low fertility soils: A review of current status, and future prospects. Geoderma. 2019, 337, 536-554. https://doi.org/10.1016/j.geoderma.2018.09.034.
- Ghosh, D.; Masto, R. E.; Maiti, S. K. Ameliorative effect of Lantana camara biochar on coal mine spoil and growth of maize (Zea mays). Soil Use Manag. 2020,36(4), 726-739. https://doi.org/10.1111/sum.12626
Table 4. Physico-chemical characteristics of soil having different application rate of rice straw derived biochar (RS-BC) and bone char (WB-BC) alone and in combination (RS-BC+WB-BC) after 15 days incubation period. Values are means of three replicates ± SD
|
Treatments |
Soil Parameters |
||||||||
|
Application rate |
pH |
ECe (µS cm-1) |
OM (%) |
WHC (%) |
TDS (ppm) |
CEC (cmolc kg-1) |
BD (g cm-3) |
SOC (%) |
|
|
Control |
|
7.95CD±0.10 |
104.46EF±6.30 |
3.13F±0.03 |
28.33D±5.68 |
118.8CD±16.99 |
8.33E±0.29 |
1.40B±0.05 |
3.56CD±0.20 |
|
CF |
|
8.19BCD±0.04 |
161.63B±3.90 |
3.48E±0.04 |
27.66D±4.04 |
139.46BC±6.55 |
9.2E±0.38 |
1.58A±0.11 |
4.35BC±0.41 |
|
RS-BC |
5% |
7.60cE±0.22 |
108.4cE±6.10 |
3.41cE±0.11 |
32cC±6.24 |
135.8aBC±6.62 |
9.6bE±0.35 |
1.28aCD±0.02 |
4.73cB±0.17 |
|
10% |
7.93bCD±0.1 |
114.5bD±5.86 |
3.59bD±0.15 |
55.6aB±8.02 |
108.2bD±3.15 |
11abD±0.57 |
1.14bDE±0.03 |
5.30bB±0.17 |
|
|
15% |
8.25aBC±0.1 |
131.8aC±3.23 |
3.94aCD±0.08 |
67aA±2.64 |
145.3bD±10.25 |
12.5aC±0.37 |
1.02cE±0.02 |
6.26aA±0.37 |
|
|
WB-BC |
5% |
7.9bDE±0.32 |
109.3cE±12.09 |
3.64bD±0.20 |
30.3bCD±6.02 |
81.83cE±7.91 |
11.1cD±0.18 |
1.35aC±0.04 |
3.16bD±0.11 |
|
10% |
8.1bBCD ±0.17 |
117.8bD±20.12 |
3.98bBC±0.12 |
50.66aBC±7.7 |
109.6bD±18.36 |
12.9bBC±0.52 |
1.24bCD±0.03 |
4.58aBC±1.69 |
|
|
15% |
8.4aB±0.18 |
183.1aA±23.23 |
4.40aA±0.47 |
63.66aAB±6.8 |
169.7aA±12.09 |
14.6aA±0.44 |
1.12cDE±0.02 |
4.73aB±0.18 |
|
|
RS-BC+WB-BC |
5% |
8.12cBCD±0.1 |
124.2cCD±8.67 |
3.93bCD±0.20 |
34.33bC±3.05 |
146.66aB±17.77 |
11.9bCD±0.54 |
1.36aC±0.04 |
4.99aB±0.18 |
|
10% |
8.42bB±0.09 |
127.06bC±9.32 |
4.22abBC±0.1 |
53.40abB±5.10 |
107.06bD±7.47 |
14aAB±0.57 |
1.28bCD±0.02 |
4.89aB±0.22 |
|
|
15% |
8.9aA±0.22 |
171.73aA±9.86 |
4.33aA±0.20 |
66.33aA±3.21 |
136aBC±12.61 |
15.33aA±0.4 |
1.21bD±0.01 |
5.24aB±0.19 |
|
Table 5. Physico-chemical characteristics of post-harvested soil after harvest of 65 days old ridge gourd plants. Values are means of three replicates ± SD
|
Treatments |
Soil Parameters |
||||||||
|
Application rate |
pH |
ECe (µS cm-1) |
OM (%) |
WHC (%) |
TDS (ppm) |
CEC (cmolc kg-1) |
BD (g cm-3) |
SOC (%) |
|
|
Control |
|
7.95C±0.05 |
143.1F±2.73 |
3.74CD±0.11 |
28.83F±1.04 |
126.43H±3.68 |
7.9G±0.16 |
1.38A±0.01 |
3.47F±0.07 |
|
CF |
|
8.1C±0.06 |
155.7E±4.98 |
3.75CD±0.05 |
30.66E±2.51 |
140.6G±4.46 |
8.8FG±25 |
1.32AB±0.02 |
4.67E±0.15 |
|
RS-BC |
5% |
7.66cD±0.10 |
118.2cH±2.56 |
3.64bD±0.10 |
35.66cD±1.15 |
139.9bG±3.07 |
9.5bF±0.34 |
1.14aBC±0.01 |
6.43cD±1.28 |
|
10% |
7.94bC±0.10 |
130.8bG±3.81 |
3.77abCD±0.04 |
38bD±2.64 |
145.33bFG±4.75 |
10.9abE±0.52 |
1.12bFG±0.02 |
6.56bCD±1.25 |
|
|
15% |
8.21aBC±0.07 |
145aF±3.2 |
3.92aC±0.08 |
58.3aC±3.51 |
171.66aE±6.50 |
12.3aCD±0.35 |
1.10cC±0.01 |
7.81aC±0.54 |
|
|
WB-BC |
5% |
7.56bD±0.18 |
159.33cE±4.04 |
3.98bC±0.23 |
56.33bC±1.52 |
126.16cH±4.85 |
11.4bDE±0.31 |
1.21aB±0.03 |
6.48cD±1.28 |
|
10% |
8.20aBC±0.05 |
171.5bD±3.77 |
4.65aB±0.15 |
62.93aBC±2.61 |
153.73bF±5.93 |
12.80bC±0.5 |
1.18bB±0.01 |
8.01bD±0.57 |
|
|
15% |
8.21aBC±0.03 |
199.26aC±5.21 |
4.71aB±0.18 |
63.66aBC±0.57 |
189.7aD±5.01 |
15aAB±0.23 |
1.16cBC±0.02 |
8.58aA±0.38 |
|
|
RS-BC+WB-BC |
5% |
8.43aAB±0.07 |
241.8cB±5.38 |
4.69bB±0.12 |
63.66bBC±1.52 |
226.73cC±9.40 |
12bCDE±0.31 |
1.18aB±0.03 |
7.81cC±0.17 |
|
10% |
8.49aA±0.39 |
246.26bB±15.86 |
4.84bB±0.08 |
69.75aAB±0.43 |
315.16bA±10 |
14.6aB±0.30 |
1.08bCD±0.01 |
8.08bB±0.52 |
|
|
15% |
8.67aA±0.09 |
340.33aA±5.68 |
5.26aA±0.25 |
74.33aA±0.57 |
291.33aB±4.04 |
16.1aA±0.21 |
1.03bD±0.02 |
8.64aA±0.25 |
|